# Text2Decision: Decoding Latent Variables in Risky Decision Making from Think Aloud Text

## Abstract

Understanding human thoughts can be difficult, as scientists usually rely on observing behaviors. The Think-Aloud protocol, where people talk about their thoughts while making decisions, provides a more direct way to study thoughts. However, past research on this topic has mostly been qualitative. Recent advancements in artificial intelligence and natural language processing provide the potential for more quantitative analysis of language data. This study introduces **Text2Decision**, a model trained on task questions from a large-scale task collection, used to decode decision tendencies in risky decision-making from Think-Aloud texts. We test our model in both human and GPT-4 simulated Think-Aloud text data about risky decision-making, which are out-of-distribution in the training. Our findings demonstrate the model's performance in capturing GPT-4 manipulated decision personas and in unveiling heuristic decision tendencies from humans. **Text2Decision** demonstrates its capability by training on basic task outlines and theoretical frameworks and generalizing to unseen empirical Think-Aloud text data. This not only allows decoding individual differences from these texts but also extends to analyzing large-scale domain datasets. This study shed light on AI integration in cognitive research for the AI4Science paradigm.

## 1 Introduction

Understanding human thoughts is one of the major goals of Cognitive Science, yet observing thoughts is hard. Inspired by the behaviorist approach, modern Computational Cognitive Scientists have tried to infer hidden thought processes by fitting computational models to behavioral data that is easy to observe — such as button presses and response times. While this approach has had some notable successes, it also suffers from several limitations, not least of which is that the models' design is often colored by the researcher's own cognitive experiences and introspections[Wilson and Collins, 2019].

One of the more direct methodologies to access human thoughts is to simply ask people to speak them aloud, via the Think-Aloud procedure [Simon and Ericsson, 1984]. However, due to the complexities and intricacies of linguistic data, traditional analyses of Think Aloud data have been limited by human coding capacities to be largely qualitative and relatively small scale. As an example of this traditional approach, and of the type of task we will later use in this paper, Brandstätter and Gussmack [2013] used the Think-Aloud procedure in risky decision-making tasks. In this study, the researchers hand-coded people's utterances according to which kind of decision-making strategy they might be using and whether this was closer to a holistic strategy like Prospect Theory, where options in

Submitted to NeurIPS 2023 AI for Science Workshop.

a choice problem are assigned a single Expected Utility [Kahneman, 2011], or a heuristic strategy, where the features of options are compared one by one [Gigerenzer and Gaissmaier, 2011].

While this study provided support for the heuristic view of decision-making, the hand-coding approach was time-consuming, subjective, and hard to replicate, all of which challenge the deeper application of the Think-Aloud method[Gu, 2014].

The contemporary landscape of artificial intelligence, notably in the realm of natural language processing (NLP), has brought about transformative changes. Large language models (LLMs) are now equipped to process, understand, and even reason with language data, thereby offering the potential to bridge the gap between qualitative nuances and quantitative rigor [Zhao et al., 2023]. With these advances, we have the opportunity to process Think-Aloud text data with advanced quantitative techniques, decoding latent interpretable variables directly from Think-Aloud text, without the intervention of human coding.

In our present study, we introduce the **Text2Decision** neural network model. This model is designed to decode both overt human behaviors and the subtle, underlying variables at play during risky decision-making tasks, all from Think-Aloud text narratives.

## 2 Risky decision-making task

To help ground our exposition of the **Text2Decision** model, we first introduce the cognitive task that is the focus of this paper. As our first foray into the Think-Aloud paradigm, we sought to replicate the decision-making study of Brandstätter and Gussmack [2013]. This study combines 18 decision problems from Kahneman and Tversky's classic studies [Kahneman and Tversky, 1979] with the Think-Aloud procedure. For each decision problem, participants choose between two gambles, each offering varying outcome-probability combinations (e.g. $10 for sure or $20 with 50% chance). We had both humans and artificial agents complete the task.

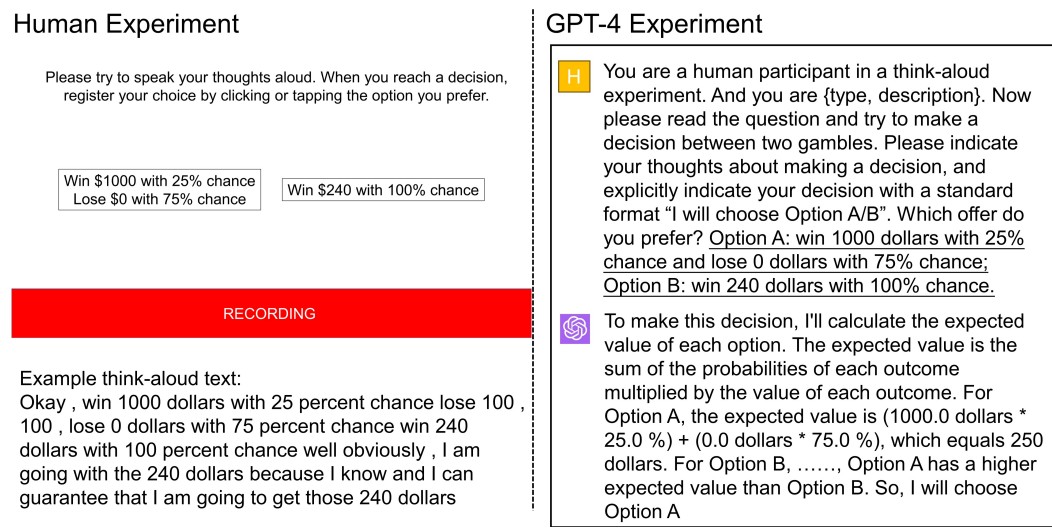

Figure 1: Risky Think-Aloud decision-making task run on humans and GPT-4

### 2.1 Human experiment

76 undergraduate students recruited from the University of Arizona Psychology Subject Pool completed the task online recording both their choices and verbal utterances of their thoughts. Transcripts

were initially generated using OpenAI's Whisper [OpenAI, 2023b] and subsequently verified by human research assistants. The study was approved by the University of Arizona IRB.

## 2.2 GPT-4 experiment

To acquire a Think-Aloud and behavioral dataset from GPT-4, we instruct GPT-4 to Think-Aloud and decide as if it were a human participant with the exact same trials in the human experiment. Simulating the same sample size (N=76), we also employed GPT-4 and attributed it to one of five decision-making personas: 'Risk-Averse', 'Risk-Seeking', 'Rational Decision-Maker', 'Probability-Weighted Decision-Maker', or 'Outcome-Focused Decision-Maker'. The specifics of these personas and instruction prompts are detailed in the Appendix Section A. By manipulating the prompts, GPT-4 was instructed to generate varied styles of Think-Aloud responses. For decisions, GPT-4 was guided to use a consistent format, "I will choose A/B", enabling us to extract choices using regular expressions.

# 3 Text2Decision model

To decode interpretable latent decision variables from Think-Aloud data, we need a model that bridges between semantic and theory-driven domains. It is essential that this model be robust and versatile, able to capture varied latent variables across distinct data patterns. While many models require substantial empirical datasets, our **Text2Decision** framework is designed to train efficiently on easily generated synthetic datasets and seek interpretations on empirical Think-Aloud datasets.

## 3.1 Model

The **Text2Decision** model is a fully connected neural network that translates text into a compact set of pre-defined decision variables. The goal is that applying the model to Think-Aloud data will allow us to extract the decision variables used by the agent (human or GPT-4) to make its decision. In the risky decision-making task, these decision variables include key features of the gambles that are thought to drive decisions such as the expected values, probabilities, losses, and gains associated with each gamble.

To train the network to perform this transformation from text to decision variables, we made use of a large collection of 14,568 risky decision problems used in Peterson et al. [2021]. Our basic approach was to input the text of the question and train the network to predict the vector of decision variables we computed from the decision problem.

More concretely, inputs to the network took the form of text embeddings produced using OpenAI's *text_embedding_ada_002* model, based on the mere question descriptions[OpenAI, 2023a]. For example, for the question used in Figure 2, the input text would be '1000 dollars with $50.0\%$ chance, 0 dollars with $50.0\%$ chance.'.

The decision variables to be predicted by the network included both heuristic and normative measures that have been hypothesized to drive human decision-making Kahneman [2011]. Heuristic decision variables included **maximum gain, minimum gain, maximum loss, minimum loss, maximum plus median gain, probability of maximum gain, probability of minimum gain, probability of maximum loss, probability of minimum loss, probability of maximum plus median gain** [Brandstätter and Gussmack, 2013]. Normative decision variables included **Expected Utility** and **Entropy**. Thus, for the example decision used in Figure 2, the decision variable vector would be [1000, 0, 0, 0, 1000, 0.5, 0.5, 0,0, 1, 500, 1].

The model's objective is to learn the correlation between text and decision variables, decoding pertinent Think-Aloud text into interpretable decision variables, which could be further investigated in clustering individual differences or uncovering algorithmic information in decision-making.

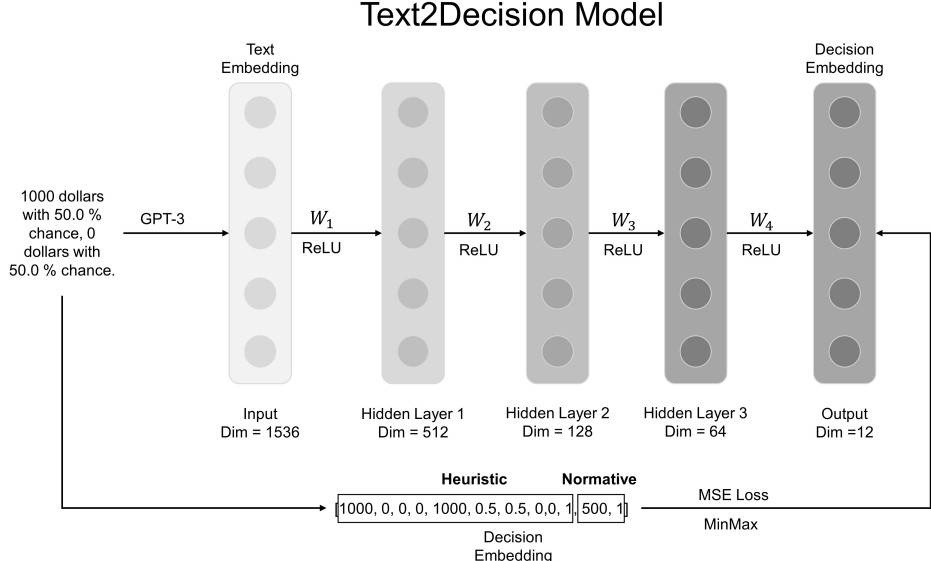

Figure 2: Text2Decision model structure and training illustration

 ### 3.2 Training and validation

For each question in the collection, we generated corresponding text and decision variables. The dataset is partitioned into $80\%/10\%/10\%$ segments for training, validation, and testing, respectively. Our loss function is the Mean Squared Error (MSE), defined for a 12-dimensional decision variable as:

$$\text{MSE} = \frac{1}{12} \sum_{d=1}^{12} \sum_{i=1}^{N} (y_{i,d} - \hat{y}_{i,d})^2$$

Considering the varied scales across the decision variable dimensions (e.g., probabilities between 0 and 1 versus outcomes from -3000 to 3000), we employed min-max normalization for each dimension prior to training, which ensures a balanced training impact. Using the Adam optimizer with batch gradient descent, we minimize the MSE across 200 epochs at a learning rate of 0.001. Both training and validation losses are monitored per dimension, verifying thorough model transformations (Figure 3). Validation outcomes underscore the model's proficiency in computing decision variables from question descriptions.

## 4 Results

Next, we asked whether transforming Think-Aloud text to decision variables via the **Text2Decision** network enabled us to better predict choices of both humans and GPT-4.

To predict choices from Think-Aloud texts, we devised four logistic regression models to assess if our **Text2Decision** model offers enhanced predictive performance relative to basic text embeddings from GPT-3.

In particular, we tested three different flavors of **Text2Decision**. In the first, 'Text2Decision embedding,' we used the decoded decision variables as inputs to the logistic model. In the second, 'Text2Decision relative Euclidean distance,' we compared the decoded decision variables from the Think-Aloud text to the decoded variables generated from each of the two options. The idea here is that the closest option to the text is more likely to be chosen. In the third flavor of the model,

## Training & Validation

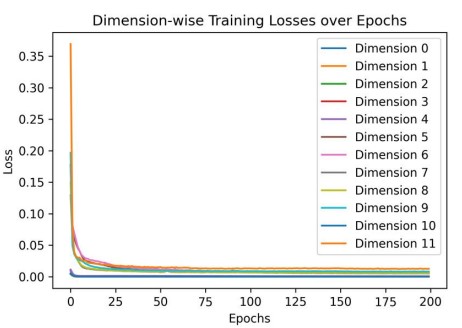
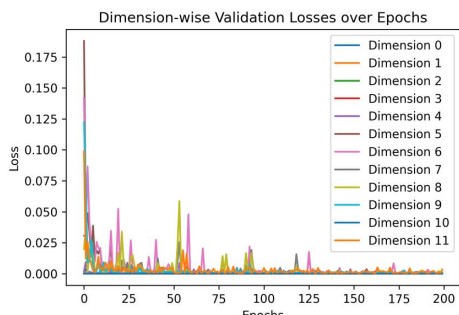

Figure 3: Text2Decision model training and validation loss

Table 1: Decision prediction performance of logistic regression models

| Model Name | Mean Accuracy ($\pm$ SE) | |
| --- | --- | --- |
| | GPT-4 | Human |
| Raw embedding (PCA to 12 Dimensions) | $0.69 \pm 0.01$ | $0.55 \pm 0.01$ |
| Text2Decision embedding | $0.58 \pm 0.01$ | $0.62 \pm 0.01$ |
| Text2Decision relative Euclidean distance[a] | $0.59 \pm 0.01$ | $0.52 \pm 0.01$ |
| Text2Decision multi-dimensional relative Euclidean distance[a] | $0.65 \pm 0.01$ | $0.68 \pm 0.01$ |

[a] The relative distance means the differences in distances from the Text2Decision model transformed text embedding to decision embeddings of two given options.

'Text2Decision multi-dimensional relative Euclidean distance,' we adopted a similar idea of comparing the Euclidean distance of decoded decision variables from the Think-Aloud text to the decoded variables generated from each of the two options but leaving them separately computed for each dimension as a per regressor in the logistic model. Finally, as a baseline, we trained a model based on the raw text embeddings compressed to 12 dimensions by principal component analysis, 'Raw embedding.'

All models we evaluated based on Leave-One-Out-Cross-Validation (LOOCV). Results of this analysis are shown in Table 1 for both the GPT-4 and human Think-Aloud data.

For the GPT-4 dataset, the baseline model (Raw embedding) is the most accurate in predicting agents' choices.[1] For the human dataset, the model leveraging multi-dimensional relative distances between transformed text embedding and the two provided options (Text2Decision multi-dimensional relative Euclidean distance) excels over its counterparts. This suggests that our Text2Decision model can effectively predict choices based on Think-Aloud text.

### 4.1 Decoding manipulated personas in GPT-4

A key application of our model is to decode latent interpretable variables. In the GPT-4 experiment, we introduced varied decision-maker types to assess the model's capability in discerning individual differences in risky decision-making. Using **Text2Decision**, we transformed each text embedding into the decision variables and computed the variance for each dimension across the five decision-maker

---

[1] In fact, we achieved near-perfect accuracy of 99% with GPT-4's original output. This likely stems from GPT-4's consistent phrasing, such as 'I will choose A/B', potentially being deterministically encoded into the text embedding. However, when we masked decisions by substituting A/B with X, performance significantly declined. We masked the decision information because we want to ensure the model learns the capacity of inference from the Think-Aloud reasoning process, but not keywords of decisions in a statistical pattern reflected in text embedding.

types. Given that all decision-makers underwent identical trials, we attribute variance differences among them to individual disparities.

As depicted in Figure 4, the five decision-maker types display varied variances across certain dimensions. Notably, the 'Probability-Weighted Decision Maker' type manifests the largest variances in 'Expected Utility', 'maximum loss', and 'probability of minimum loss'. This aligns with its descriptive prompt: 'Relies on explicit probabilities to estimate expected values and opts for choices with the highest perceived value.' For a comprehensive description of all types, refer to the Appendix (see Section A).

These findings indicate that the transformed text embeddings from Think-Aloud data can unveil latent variables in risky decision-making tasks, aligning with the ground truth from our GPT-4 manipulations. Consequently, this method holds promise for hypothesis testing in experimental settings.

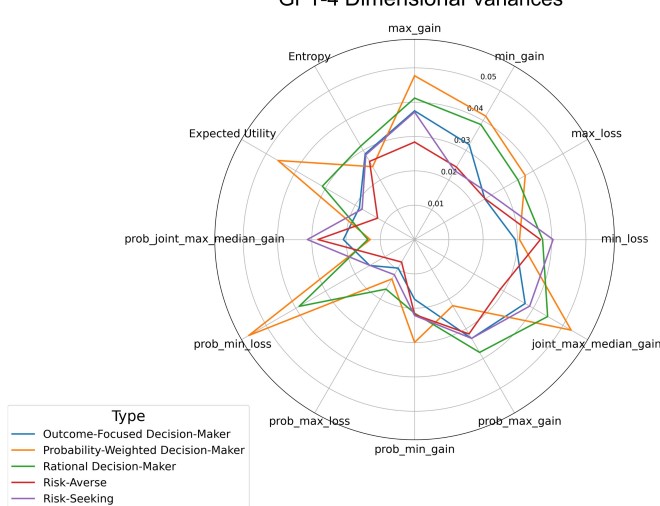

Figure 4: Decoding individual differences in GPT-4 data by computing variance for each dimension of Text2Decison transformed Think-Aloud embedding

## 4.2 Decoding individual differences in humans

In a similar vein, we sought to decode latent variables from human data. Without explicit labels for categorization in this dataset, we employed K-means to cluster and segment individuals into five categories. Variance calculations for each label's participants are illustrated in Figure 5.

Distinctive decision-making styles emerge for each participant cluster. For instance, Cluster 0 participants show the highest variance in 'probability of maximum loss' and 'probability of minimum loss', suggesting a loss-focused approach. Conversely, Cluster 3 participants exhibit pronounced variance in 'probability of max gain' and 'probability of maximum plus median gain', indicating a gain-centric perspective with a probability over outcome emphasis. Interestingly, no clusters display significant variance in the Expected Utility dimension. This hints at humans being more heuristic-driven in their decision-making for this task, in contrast to the normative-driven GPT-4. The heuristic nature of human risky decision-making revealed by the variance representations in Text2Decision transformed embeddings aligns with the findings in behavioral experiments [Kahneman and Tversky, 1979] and Think-Aloud experiments with hand-coding[Brandstätter and Gussmack, 2013].

Integrating unsupervised learning methods with our **Text2Decision** model's output, we present a framework that allows for interpretable insights into clusters, labels, or principal components.

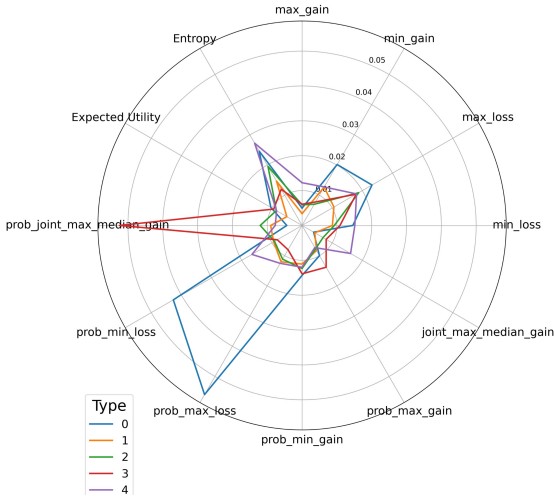

Figure 5: Decoding individual differences in human data by computing variance for each dimension of Text2Decison transformed Think-Aloud embedding

## 5 Discussion

In this research, we introduced the **Text2Decision** framework model designed for decoding latent variables from Think-Aloud texts. We assessed the model's efficacy in behavioral prediction and discerning individual differences. Our findings indicate that the model adeptly extracts pertinent information from raw Think-Aloud text embeddings, enhancing behavioral predictions, particularly with human datasets. Moreover, the model demonstrated the capability to decode latent, interpretable variables.

Our framework seeks to bridge the gap between hypothesis-driven and data-driven methodologies in the Cognitive Sciences by integrating the Think-Aloud protocol with Large Language Models. Historically, Cognitive Scientists have depended on experimental manipulations to validate hypotheses. With the advent of computational modeling, there's been a shift towards quantitatively characterizing behaviors to provide generative explanations. Yet, a persistent challenge is navigating the trade-off between the interpretability inherent in hypothesis-driven approaches and the precision of data-driven ones. Unlike models anchored to specific hypotheses, our **Text2Decision** framework is trained on general task settings, ensuring a broader explanatory capacity on empirical data, accommodating the potential for both hypotheses testing and interpretable data-driven investigations. By mapping semantic spaces to interpretable decision spaces, we aim to decode latent variables, making them comprehensible and primed for further exploration.

Moving forward, our aim is to both deepen and broaden the **Text2Decision** framework. In terms of depth, we plan to integrate computational modeling, behavioral analysis, and neural recordings (e.g., fMRI or EEG) to facilitate more robust hypothesis testing and extract clearer, interpretable patterns[Schneider et al., 2023]. Moreover, we intend to explore diverse participant populations, considering factors such as race, gender, culture, age, and mental health, to better understand their Think-Aloud representations during risky decision-making. Broadening our scope, we aspire to adapt our framework to more intricate tasks, including learning, planning, and challenges like sorting, clustering, and compositionality. These tasks usually contain rich slow cognitive processes, whereas Think-Aloud texts may be more useful to decode complex strategies and algorithms. We are also keen to assess whether our strategy of training on basic task settings (or easily generated synthetic data) retains its efficacy in these diverse contexts.

In conclusion, our framework, in tandem with LLMs, heralds a promising avenue for deciphering human thought processes via Think-Aloud methodologies.

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

# A  GPT-4 Experiment Prompts

"You are a human participant in a Think-Aloud experiment. And you are {type, description}. Now please read the question and try to make a decision between two gambles. Please indicate your thoughts about making a decision, and explicitly indicate your decision with a standard format 'I will choose Option A/B'. Which offer do you prefer? Option A: win 1000 dollars with 25% chance and lose 0 dollars with 75% chance; Option B: win 240 dollars with 100% chance."

## A.1  Types of Decision Makers

1. **Risk-Averse:**
   Prefers options with predictable outcomes and minimal risk, even if potential rewards are lower.

2. **Risk-Seeking:**
   Drawn to high-reward options even if they come with significant risks.

3. **Rational Decision-Maker:**
   Analyzes all available information and weighs pros and cons to maximize the outcome.

4. **Probability-Weighted Decision-Maker:**
   Relies on explicit probabilities to estimate expected values and chooses options with the highest perceived value.

5. **Outcome-Focused Decision-Maker:**
   Prioritizes potential outcomes, especially extreme values, and may avoid options with possible losses even if the expected value is positive.

