# OpenReview forum: "Text2Decision: Decoding Latent Variables in Risky Decision Making from Think Aloud Text"
_NeurIPS.cc/2023/Workshop/AI4Science — NeurIPS2023-AI4Science Poster_

### Official Review · Reviewer_HvAf · 2023-10-20
**title_placeholder**

**Rating:** 6
**Confidence:** 3

**Review:**

Summary

This papers proposes Text2Decision, a simple neural network trained to predict decision variables froms think-aloud text. Using Text2Decision, the authors analyzed how both human and GPT-4 makes decisions and made a few interesting observations in a quantitative manner.


Strength
- Interesting observation of the differing focus between human and GPT-4's decision variables, e.g., GPT-4 can be prompted with different personas that employ different latent decision variables to make a decision, humans rely more on heuristic for decision making.


Weakness
- Clarify could be improved.
- Interpretation of results is sometimes missing. See details in additional comments.
- Lack of novelty w.r.t. to the method and sub-optimal performance, e.g., performs worse than simple baseline on GPT-4 generated think-aloud text.


Additional Comments
- A naive baseline is simply prompt GPT-3.5/4 to output the decision making variables from text. I'm curious how it might fair compared to Text2Decision model.
- Please clarify the following in writing! How are GPT-4 answers simulated to match human trial sample size? Perhaps via repeated calls to openai api? Is the variance computed in Figure 4 over this simulated samples (N=76) for each of the personas?
- The authors would benefit from clarifying what "text embedding" is referring to exactly, e.g., Line 142. It seems text2decision model takes text embedding for one out of the two options in the decision making problem for training (seen in Figure 2.), but during inference the text2decision model takes text embedding of think-aloud text generated by either human or GPT-4. Is this true?
- In Section 4.1, why is there a higher variance in the predicted decision variables for "probability-weighted decision maker" persona? Explain a bit in text.
- In section 4.2, why is there no explicit labels for categorization in the human data? Clarify that you could not separate human into sub-groups with different levels of risk-adverseness.
- In Section 4.2, mention what is being clustered.

---

### Meta-Review · Area_Chair_Apgy · 2023-10-26

**Recommendation:** Accept (Poster)
**Confidence:** 3

**Metareview:**

The paper present a natural and intruiging use of LLMs in social sciences to process verbalized human thinking. The reviewer appreciated the findings, while pointed out some noncritical issues, e.g. with clarity. Thank you for your submission and I am happy to recommend acceptance of the paper.